# Unsupervised Indoor Positioning System Based on Environmental Signatures

**DOI:** 10.3390/e21030327

**Published:** 2019-03-26

**Authors:** Pan Feng, Danyang Qin, Min Zhao, Ruolin Guo, Teklu Merhawit Berhane

**Affiliations:** 1Key Lab of Electronic and Communication Engineering, Heilongjiang University, Harbin 150080, China; 2Department of Computer and Information Sciences, Dire-Dawa Institute of Technology, Dire Dawa 3000, Ethiopia

**Keywords:** indoor positioning, unsupervised positioning, SLAM, mobile sensor

## Abstract

Mobile sensors are widely used in indoor positioning in recent years, but most methods require cumbersome calibration for precise positioning results, thus the paper proposes a new unsupervised indoor positioning (UIP) without cumbersome calibration. UIP takes advantage of environment features in indoor environments, as some indoor locations have their signatures. UIP considers these signatures as the landmarks, and combines dead reckoning with them in a simultaneous localization and mapping (SLAM) frame to reduce positioning errors and convergence time. The test results prove that the system can achieve accurate indoor positioning, which highlights its prospect as an unconventional method of indoor positioning.

## 1. Introduction

In outdoor environments, the Global Navigation Satellite System (GNSS) can provide satisfactory positioning, but satellite signals are blocked by walls, causing no signal in the rooms. At the same time, as the indoor environments become more complex, and indoor activities become more frequent, the demand for indoor positioning is increasingly strong.

In order to achieve better indoor positioning results, a lot of research has been carried out. Many papers focus on WiFi indoor positioning based on received signal strength, a common local positioning approach with a number of prominent advantages such as low cost and ease of deployment. Ref. [1] proposed an indoor positioning method based on a Monte Carlo algorithm. Ref. [2] proposed the improved indoor position estimation algorithm based on geo-magnetism intensity, which compensates for the flaw of particle filter. Ref. [3] also presented a novel method, and it proposed a low-cost and easy-to-realize positioning system, which uses FM Radio Signal Strength fingerprinting.

Many of today’s localization systems for indoor and outdoor positioning are based on propagation time measurements of radio signals. Ref. [4] proposed a new WiFi-based positioning method to make the access point (AP) signal features more complete. Ref. [5] proposed two indoor positioning algorithms based on channel state information (CSI). Ref. [6] proposed a new fingerprint-based WiFi indoor positioning system, which is realized by extracting and analyzing the individual multipath propagation delay.

However, in order to achieve high positioning accuracy in the presence of non line of sight (NLOS) propagation, these systems require either an expensive manual calibration or additional information. Many paper introduced unsupervised positioning without these costs. Ref. [7] proposed a Wi-Fi radio maps (WRMs) calibration system that automates the initial construction and maintenance of radio maps using crowdsourced fingerprints collected from numerous smartphones without location information. The system incorporates an unsupervised learning algorithm into an incremental and adaptive calibration process. Ref. [8] proposed an unsupervised indoor localization scheme that uses the combination of smartphone sensors, iBeacons and Wi-Fi fingerprints for reliable and accurate indoor localization with zero labor cost. Ref. [9] proposed a novel approach for a channel impulse response (CIR) based fingerprint system, which reduces the calibration and measurement effort and simultaneously improves localization results. Inspired by this, the paper designed an unsupervised indoor positioning system to reduce development costs and improve positioning accuracy.

Although there are many works about indoor positioning, there are still some problems: some indoor positioning technologies rely on customized installations, which reduces their scalability [10,11]. The positioning systems based on WiFi [12,13], while providing ubiquitous positioning, require cumbersome calibration works. Some systems can reduce the works [14,15,16], but most need to reduce positioning accuracy. The dead reckoning approach based on sensors of smartphones has begun to receive attention [17,18] in recent years, which uses an accelerometer to calculate the displacement of the user and a compass to learn the orientation of the user. Ref. [19] proposed a method of indoor navigation using a microelectromechanical systems (MEMS)-based strapdown inertial navigation system (INS) aided by Wi-Fi signal strength measurements. Ref. [20] proposed an algorithm to detect the stride using acceleration spectrogram feature by utilizing the accelerometer in a smartphone. The dead reckoning error is quickly accumulated, so the user position needs to be recalibrated.

The paper proposes an unsupervised indoor positioning (UIP) system, using sensors on smartphones to detect unique indoor points, such a place where a unique group WiFi APs can be received, which can reset the errors in dead reckoning. Starting with the building layout generated manually or automatically [21,22,23], the system discovers the landmarks through the crowd-sensing method. The landmarks are then used to reset the errors in the dead reckoning, resulting in higher positioning accuracy. The test results prove that the system can achieve accurate indoor positioning, showing the prospect of UIP as an unconventional method of indoor positioning.

The rest of this paper is organized as follows: Section 1 introduces the shortcomings of existing indoor positioning research and the research done in this paper. Section 2 introduces how the system uses sensor data to get the displacement and orientation of the user, and proposes a novel method by using landmark in the positioning process. Section 3 describes how to define landmarks by sensor data features and how to identify different landmarks in the experiment, and it is an important innovation in the paper. Section 4 introduces simultaneous localization and mapping and how it works in positioning systems. Section 5 tests the system UIP from different aspects, and the results prove that it has excellent positioning performance. Section 6 provides the conclusions.

## 2. Related Work

There are also two representative indoor positioning systems: the Horus system and the MaLoc system, and they are compared with the UIP system in the test. The Horus system identifies different causes for the wireless channel variations and addresses them to achieve its high accuracy. It uses location-clustering techniques to reduce the computational requirements of the algorithm. The Horus system works in two phases. The offline phase builds the radio map, clusters radio map locations, and conducts other preprocessing of the signal strength models. The online phase estimates the user location based on the received signal strength from each access point and the radio map prepared in the offline phase.

The MaLoc is built on a proposed augmented particle filter. To minimize errors in motion estimation and improve the robustness of particle filter, it augments the particle filter with a dynamic step length estimation algorithm and a heuristic particle resampling algorithm. It uses a hybrid measurement model which combines a new magnetic fingerprinting model and the existing magnitude fingerprinting model to improve the system performance and avoid calibrating different smartphone magnetometers.

This paper is inspired by these two methods, using a probability calculation method similar to the Horus system, and using the magnetic signal as the positioning technology, which is the method of the MaLoc system. At the same time, this paper has made improvements to the two systems. The main improvements are as follows:The paper uses multi-domain sensors to make up for the single signal to be easily affected by the environment, and the positioning accuracy is insufficient. For example, Horus system is susceptible to noise in the channel.The paper proposes global landmarks, which has not been seen in past positioning systems. Global landmarks enable the system to obtain certain environmental information in advance, which helps to improve positioning accuracy.This paper makes use of the simultaneous localization and mapping (SLAM) architecture and framework that leverages smart phone sensors to both dead-reckon the user location and identify semantic landmarks. These landmarks are used in a SLAM probabilistic framework to reset the accumulated localization error. It makes the UIP system have high positioning accuracy without initial manual calibration.

## 3. Mobile Sensor Positioning

### 3.1. Dead Reckoning

Dead reckoning is the method to estimate the track of the users and it needs to sample the user’s motion data (e.g., acceleration). The algorithm can calculate the possible displacement and direction of the user from these data, and finally calculate the possible position of the user at the next sampling. In the mobile device positioning system, accelerometer and compass are used to get acceleration and orientation, then process and calculate these data to get the possible position of the user at the next time. Starting from a point which is known, the user’s position at time *t* is updated by the control variable $u_{t}$ = $\{l_{t}$, $\phi_{t}\}$, where $l_{t}$ is the displacement and $\phi_{t}$ is the orientation change.
(1)Displacement: It can be calculated simply by the direct integral of the accelerometer reading which can be obtained by acceleratometer. However, it has a bad positioning error. As shown in Figure 1, after the actual displacement of 30 m, the deviation of the estimated value exceeds 100 m; this is due to the noise and low sampling rate of the accelerometer and the jitter of the phone when the user is walking. The paper uses another effective method.It is identifying the feature of the user’s walking [18,24]. The feature comes from the natural upward/downward rebound of the body at every step. To capture it, the accelerometer signal is processed by a low-pass filter, and the system identifies two consecutive local minimum, if the difference value between maximum and minimum is larger than the given threshold, it will be considered as a step of the user, and the displacement is calculated by multiplying the steps by the step size.(2)Orientation: Traditional approaches rely on the compass, which can indicate the orientation by the magnetic field. However, it is noisy and seriously impacts positioning results. Gyroscope is also used in the paper, and it can measure relative palstance, which is a three-dimensional rotation matrix, and the relative angular displacement (RAD) can be obtained when the matrix is multiplied by a time interval. The path structure of the user can be tracked by the gyroscope; however, the estimated path has an error in the initial orientation. As shown in Figure 2a, all of these paths are rotated versions of the real path.Landmarks can correct the deviation. As shown in Figure 2b, the position X${}_{1}$ of the user at time $t_{1}$ is known, dead reckoning with initial bias θ estimates that the position is X${}_{3}$ at time $t_{2}$, and the real position is X${}_{2}$. Assuming that the user encounters the landmark L${}_{2}$ at time $t_{2}$ and the location of L${}_{2}$ is X${}_{2}$, which is known, the algorithm thus considers that the the location of the user is X${}_{2}$, which is an accuracy position.

### 3.2. Landmark Introduction

#### 3.2.1. Landmark Density Indoors

According to observations, signals in indoor environments are abundant, such as sound, light, and magnetic field. In addition, some building structures force the user to move in specific ways, which might create some unique patterns on the sensors. Some signatures may appear after analyzing different sensor signals.

WiFi landmark: The indoor environments have many WiFi areas where all locations can receive a different set of WiFi APs, and they are in different sizes. Figure 3 shows the size distribution of WiFi AP coverage areas. If the phone can hear a group of WiFi APs in a small area, it can be considered to be in the area. The figure measures the CDF of WiFi coverage area, for example, if x is 5, y is 0.2, which means there are 20% WiFi APs, and every one of them covers an area not more than 5 m${}^{2}$. From the figure, we can know the approximate proportion of WiFi APs that are suitable as landmarks. If the area is small, the positioning error is small. There are eight and five WiFi landmarks in small areas found on the two floors of the laboratory building, and each size is less than 4 m${}^{2}$.

Magnetic/accelerometer landmark: In order to search for signatures in the magnetic sensor/accelerometer, the paper performs K-means clustering on their measurements. The members of every cluster are mapped to the corresponding physical location. The member locations in most clusters are widely dispersed in space. Fortunately, the members of some clusters are spatially tightly coupled and can be used as landmarks. Different landmarks will be combined to improve positioning accuracy and the specific method will be explained in the next section.

#### 3.2.2. Location Estimate

Suppose the system combines the different sensor data and uses k-means clustering, and discovers three sensor signatures that can be used as landmarks, as shown in Figure 4a.

Considering that dead reckoning has random error, the paper calculates position of the landmark by combining multiple dead reckoning estimates of a landmark. Intuitively, estimation errors are random and independent due to noises on the sensor and human step size. By combining these estimations with errors, it is expected that the mean will converge to a more accurate location. Figure 4b illustrates the feasibility of this method by simple calculation of centroid, where the circle represents the center of dead reckoning estimate—each one has a large error but multiple estimates with errors yield a better estimate. In order to avoid positioning errors due to approximation landmark patterns, landmarks should be in small areas and they will be identified combining with the WiFi signal. The specific method is described in the next section.

#### 3.2.3. The Effect of Regular Error Compensation

This section is added to verify the effect of landmarks in actual positioning, and the landmarks that are here are not the ones used in the system, but some simple alternatives. The way to get them is also very simple. The user walks naturally in the building with a smartphone from the entrance of the floor, and the system records readings in the inertial sensors and extracts the tuple <time, displacement, direction>. The doors and windows are marked with different numbers, and the user enters them into the system when passing through these places. These numbers are the alternatives of actual landmarks. Since the mapping is known, the actual position of the user can be known, and the effect of the landmark for the estimation error is simulated in this way. The experiment collected 10 traces starting from the entrance.

As shown in Figure 5, when pure dead-reckoning without landmark is used (black curve), the error is accumulated rapidly. The error can be corrected periodically by using the landmarks (blue curve). Although the performance is improved, the average positioning error is still about 11 m. Thus, the experiment processes the gyroscope readings to calculate the angular change during walking. Note that gyroscope readings are relative and need to be combined with the compass to infer the absolute orientation. Once the gyroscope and compass are combined (red curve), the error is significantly reduced. Adding landmark calibration on the basis (pink curve), the average error is further reduced. Figure 5 shows the effect of the landmarks, and it is necessary to further design the landmark system.

## 4. Landmark Design

This section details the landmarks of the UIP system. Landmarks refer to signals or combinations of signals in an indoor environment that can be captured by sensors and whose sensor patterns are unique in the environment so that the system can determine the location of the sensor based on landmarks. According to whether the locations can be known in advance, the landmarks are divided into global landmarks and local landmarks.

### 4.1. Local Landmark

Local landmarks (LLMs) are similar to designs in most unsupervised positioning systems. We obtain the floor graph at first and the system can learn the structure of the building and the coordinates of every point in the building. It is necessary for the radio map. In the offline phase, the user walks naturally along the corridor in the building. Since there are various signals in the indoor environment (here mainly WiFi and magnetic signals), the sensors of the mobile phone capture these signals during walking. The application on the phone can grab the sensor data and transfer it to the server for processing. Defining the signals with high levels of recognition on sensor pattern. The sensor patterns of LLMs cannot be obtained in prior and they are local features of the indoor environment, e.g., WiFi distribution in a certain area, and the phone cannot hear the same WiFi set when it leaves the area.

Due to the complexity and diversity of indoor signals, it is important to select an effective signal from a number of signals as a local landmark. In the paper, we choose WiFi and magnetic signal as landmarks because they are ubiquitous in indoor environments and the sensors of the mobile phone can easily obtain these signals. The tasks of finding local landmarks need to (1) distinguish different patterns from the sensed signals, and (2) test if the pattern is limited to a small area spatially. Figure 6 shows the operational flow, and the sensor data are collected by crowd sensing. The process can be shown in matrix form: the element <*m*,*n*> is sensor readings of the phone *m* at time *n*. The features of magnetic/inertial sensors include average, maximum, minimum, variance and average crossover, and they are media access control (MAC) ID and received signal strength indicator (RSSI) for WiFi.

The collected features are normalized, and then the clustering algorithm is performed for every sensing dimension and the combination of different signals (e.g., an accelerometer and a compass) to find the clusters having low similarity to other clusters. Given a cluster, calculate the correlation between it and other clusters. If every value is not larger than the threshold, the cluster will be a candidate. Then, test the area of the cluster. Test if all members of the cluster are in the same WiFi coverage area at first. If they do, calculate the location of each cluster member. If all members are in a small area (about 4 m${}^{2}$, the same below), the cluster is considered as a landmark.

(1) WiFi landmark: The MAC ID and RSSI are used as features. K-means clustering is applied to identify small areas with low similarity to all locations outside the regions. Assuming that a phone can receive WiFi AP sets $A_{1}$ and $A_{2}$ at locations $l_{1}$ and $l_{2}$, respectively, and the similarity *S* between $l_{1}$ and $l_{2}$ is
(1)$$S=\frac{1}{\left|A\right|}\sum_{\forall a\in A}\frac{min(f_{1}\left(a\right),f_{2}\left(a\right))}{max(f_{1}\left(a\right),f_{2}\left(a\right))},$$
where $A=A_{1}\cup A_{2}$, $f_{i}\left(a\right)$ is the RSSI of AP received by the phone at $l_{i}$, a∈A. The basic principle of the above equation is that, when the AP signal is equally strong at two locations, the similarity of the two locations is proportionally increased, and vice versa. S∈[0,1]. The paper chooses 0.4 as the threshold. Figure 7 shows the results of the test in a building in the university, and 0.4 is the point that balances the quality and quantity of WiFi landmarks.

(2) Magnetic/inertial landmarks: There are some turns in the indoor environment. Since gyroscopes provide reliable angular displacement, they have the opportunity to be used as landmarks. The feature called the bending factor is designed in the paper, which captures the concept of path curvature. Similar turns in the same WiFi area will be in the same cluster, and the algorithm will also check whether the cluster is limited to a small area when this coefficient and WiFi are integrated as a feature. Magnetic landmarks are defined by the similar process, and the magnetic field somewhere needs to be significantly different from other places and within a small area. Then, it can be considered as a landmark.

### 4.2. Global Landmark

Global landmarks (GLMs) is a novel concept, and the idea comes from thinking about the user’s own movement state: since the data from the surrounding environment can be used to calculate the coordinates of the user, the user’s own motion characteristics may also reflect the characteristics of the environment, and this data can be used as a new landmark. We naturally divide the user’s motion state into elevators, stairs, escalators, walking and static state, and these motion data can be obtained by inertial sensors. Their signal characteristics are usually unaffected by the environment, such as the user’s movement in the elevator in any building is almost the same. On the other hand, these movements often correspond to certain objects, such as elevators, and we can immediately know where they are through floor plans. These two features make them well suited for indoor positioning.

The collection process of GLMs data is the same as LLMs, and it is necessary to identify different kinds of GLMs by sensor patterns. Every one has its own pattern on magnetic sensor or inertial sensor. The following content will analyze the patterns of different GLMs.

Elevator: The typical process includes normal walking time, waiting for a period of time, walking into the elevator, overweight/weightlessness, then a stationary time period, the other overweight/weightlessness and finally leaving. As shown in Figure 8, the accelerometer exhibits distinct features. To identify elevator motion patterns, a finite state machine (FSM) is used, which relies on observed state transitions, and different thresholds are set for transition.

Escalator: After the elevator is identified, it is effortless to distinguish the escalator and the static from the walking and the stairs by the variance of the acceleration. The next step is to further distinguish the escalator from the static state, and the magnetic field’s variance can be used to identify them since the escalator is driven by the motor and the motor affects the surrounding magnetic field. As shown in Figure 9, the waveform of the static state is more stable.

Stairs: Finally, the stairs and walking need to be distinguished. When using stairs, gravity will have an obvious effect on the speed increase or decrease of the user. Compared to walking, there is higher correlation between the acceleration in the orientations of gravity and motion when users use stairs.

Landmark database is built in the offline phase, and each one has its coordinates in the radio map. The dataset of GLMs is easy to build because their location is known in advance. If we want to add a GLM (e.g., elevator) to the map, we only need to collect the required signal near each elevator and then mark it in the radio map according to the location of the elevator in the building. In contrast, the collection of landmark information requires a man to continuously walk in the experimental environment, sample various types of signals (such as user’s acceleration, environmental WiFi signal) at a fixed frequency, and upload them to the server to choose suitable ones as landmarks according to the above requirements.

## 5. Simultaneous Localization and Mapping

In an unknown environment, the user collects information from the surrounding environment (landmarks) while walking, determines the position of the surrounding by the landmark data, and corrects the coordinates of the landmark by its own position. This mode is consistent with SLAM, so the previous dead reckon and landmark system are integrated into the unified SLAM framework. This section will introduce corresponding background and the SLAM algorithm in the paper.

### 5.1. Background

#### 5.1.1. EKF-SLAM

SLAM is a concept for mobile robots [25] and the idea is used for human positioning in the paper. The system estimates the map (Θ) as well as the user’s pose (position ($x_{t},y_{t}$) and orientation ($o_{t}$)) at time *t* on a probabilistic framework:(2)$$f=p(s_{t},\Theta |u^{t},z^{t},n^{t}),$$where $u^{t}=u_{1},\ldots ,u_{t}$ is the control variable update history, $u_{t}$ is the control variable (displacement and orientation change) obtained by the sensors at time t−1. $z^{t}=z_{1},\ldots ,z_{t}$ is the user’s observation history of the environment. $n^{t}=n_{1},\ldots ,n_{t}$ are data association variables, and $n_{t}$ is the landmark’s identity observed at time *t*.

Extended Kalman filter (EKF) is a common method [26,27] which uses a high dimensional Gaussian density distribution. It decomposes $P(s_{t},\theta |u^{t},z^{t},n^{t})$ into two independent models as Equations (Equation 3) and (Equation 4), where $\theta_{n_{t}}$ is the location of the landmark $n_{t}$,
(3)$$p\left(s_{t}|u_{t},s_{t-1}\right)=h\left(u_{t},s_{t-1}\right)+\delta_{t},$$
(4)$$p\left(z_{t}|s_{t},\Theta ,n_{t}\right)=g\left(s_{t},\theta_{n_{t}}\right)+\varepsilon_{t},$$
where *h* and *g* are nonlinear functions, $\delta_{t}$ and $\varepsilon_{t}$ are Gaussian noise variables whose covariances are $R_{t}$ and $P_{t}$, respectively.

The method has two limitations, the computational complexity, which is the quadratic of the landmark number, and the data association problem, i.e., how to recognize the detected landmarks when signatures of two or more landmarks are similar.

#### 5.1.2. FastSLAM

FastSLAM combines particle filters (PF) [28,29] and EKF, and it takes advantage of the structural properties of the SLAM problem: given the motion path, the landmark estimations are independent conditionally, i.e., the correlation of the uncertainty of the landmark estimation depends only on the uncertainty of the path. When the path is known, the estimation errors of different landmarks are independent of each other.

Given the number *N* of landmarks, the path $s^{t}=\left(s_{1},\ldots ,s_{t}\right)$ is estimated as Equation (Equation 5):(5)$$ap\left(s^{t},\Theta |z^{t},u^{t},n^{t}\right)=p\left(s^{t}|z^{t},u^{t},n^{t}\right)\prod_{n=1}^{N}p\left(\theta_{n}|s^{t},z^{t},n^{t}\right).$$

Since the path of the user is unknown, FastSLAM estimates the path $s^{t}$ through the particle filter, and it uses particles to represent possible paths where each probability density function $p(\theta_{n}|s^{t},z^{t},n^{t})$ is estimated using EKF. Based on the particles, each map errors are independent conditionally. The posterior ($S_{t}^{\left[m\right]}$) of the *m*th particle contains the path $s^{t,[m]}$ and *N* landmark estimates, which is represented by the observed landmark type ${\hat{f}}_{n,t}$, mean $\mu_{n,t}^{\left[m\right]}$ and covariance $\Sigma_{n,t}^{\left[m\right]}$, as shown in Equation (Equation 6),
(6)$$S_{t}^{\left[m\right]}=s^{t,[m]},\underset{Landmark\theta_{1}}{\underset{︸}{{\hat{f}}_{1,t},\mu_{1,t}^{\left[m\right]},\Sigma_{1,t}^{\left[m\right]}}},...,\underset{Landmark\theta_{N}}{\underset{︸}{{\hat{f}}_{N,t},\mu_{N,t}^{\left[m\right]},\Sigma_{N,t}^{\left[m\right]}}}.$$

Compared to EKF-SLAM, the computational complexity of FastSLAM is logarithmically related to the number of landmarks. In addition, data association decisions can be performed on each particle, thus maintaining multiple posteriors of data associations, which is more robust [30]. FastSLAM can also handle nonlinear models, and it has been shown to converge under certain assumptions [31]. The UIP system will use the SLAM algorithm based on FastSLAM.

### 5.2. SLAM Algorithm

Like other SLAM algorithms, the algorithm involves sampling, map update, resampling and assuming there is only one landmark observed at a time to keep the algorithm generic.

#### 5.2.1. Pose Estimation

Given the control variable ${\hat{u}}_{t}$ from the dead reckoning, the measure $z_{t}$ at time *t* and the pose $s_{t-1}^{\left[m\right]}$, the pose of the particle [m] at time *t* is sampled as shown in Equation (Equation 7). All data can be obtained from the previous section, dead reckoning provides the user’s motion status, while sensor data also provides landmark observations.
(7)$$s_{t}^{\left[m\right]}∼P\left(s_{t}|s^{t-1,[m]},u^{t},z^{t},n^{t}\right).$$

Equation (Equation 7) can be expressed as a product of the pose distribution and the probability of measure $z_{t}$,
(8)$$p\left(s_{t}|s^{t-1,[m]},u^{t},z^{t},n^{t}\right)=\underset{s_{t}∼N\left(h\left(s_{t-1}^{\left[m\right]},u_{t}\right),P_{t}\right)}{\underset{︸}{p(s_{t}|s_{t-1}^{\left[m\right]},u_{t})}}\eta^{\left[m\right]}\int \underset{z_{t}∼N\left(g\left(\theta_{n_{t}},s_{t}\right),R_{t}\right)}{\underset{︸}{p(z_{t}|\theta_{n_{t}},s_{t},n_{t})}}\underset{\theta_{n_{t}}∼N\left(\mu_{n_{t},t-1}^{\left[m\right]},\Sigma_{n_{t},t-1}^{\left[m\right]}\right)}{\underset{︸}{p(\theta_{n_{t}}|s^{t-1,[m]},z^{t-1},n^{t-1})}},$$
where $P_{t}$ is the covariance matrix of the control data at time *t* and $R_{t}$ is the measure covariance at time *t*. Given the last pose $s_{t-1}^{\left[m\right]}$ and the control data $u_{t}=\left\{{\hat{l}}_{t},{\hat{\phi }}_{t}\right\}$, including estimated displacement ${\hat{l}}_{t}$ and orientation change ${\hat{\phi }}_{t}$, the pose at time *t* can be predicted as
(9)$${\hat{s}}_{t}^{\left[m\right]}∼P\left(s_{t}|s_{t-1}^{\left[m\right]},u_{t}\right).$$

Assuming that the errors satisfy Gaussian distribution, $l_{t}^{\left[m\right]}$ and $\phi_{t}^{\left[m\right]}$ satisfy following probability distributions:(10)$$l_{t}^{\left[m\right]}∼N\left({\hat{l}}_{t},\sigma_{l}\right),$$
(11)$$\phi_{t}^{\left[m\right]}∼N\left({\hat{\phi }}_{t},\sigma_{\phi }\right),$$
where $\sigma_{l}$ and $\sigma_{\phi }$ are the variances of the estimation errors. Equation (Equation 9) can be rewritten as follows:(12)$${\hat{s}}_{t}^{[m],\phi }=s_{t-1}^{[m],\phi }+\phi_{t}^{\left[m\right]},$$
(13)$${\hat{s}}_{t}^{[m],x}=s_{t-1}^{[m],x}+l_{t}^{\left[m\right]}cos\left({\hat{s}}_{t}^{[m],\phi }\right),$$
(14)$${\hat{s}}_{t}^{[m],y}=s_{t-1}^{[m],y}+l_{t}^{\left[m\right]}sin\left({\hat{s}}_{t}^{[m],\phi }\right).$$

Equations (Equation 10)–(Equation 14) are concrete implementations of the nonlinear function *h* in Equation (Equation 3).

The probability of measure $z_{t}$ involves the integration over all possible landmark locations $\theta_{n_{t}}$, and it is impossible generally. To solve this problem, the algorithm approximates *g* in Equation (Equation 3) to a linear function, resulting in the function in Equation (Equation 15).
(15)$$g\left(\theta_{n_{t}},s_{t}\right)\approx {\hat{z}}_{t}^{\left[m\right]}+G_{\theta }\left(\theta_{n_{t}}-\mu_{n_{t},t-1}^{\left[m\right]}\right)+G_{s}\left(s_{t}-{\hat{s}}_{t}^{\left[m\right]}\right),$$
where $\mu_{n_{t},t-1}^{\left[m\right]}$ is the mean of the estimated position of landmark $n_{t}$ at time *t*, and $\theta_{n_{t}}^{\left[m\right]}=\mu_{n_{t},t-1}^{\left[m\right]}$ is the estimated landmark location. $G_{\theta }$ and $G_{s}$ are the Jacobian of *g* about θ and *s*. Therefore, Equation (Equation 7) obeys the Gaussian distribution with the following parameters:(16)$$\Sigma_{s_{t}}^{\left[m\right]}={\left(G_{s}^{T}{\left(Q_{t}^{\left[m\right]}\right)}^{-1}G_{s}+P_{t}^{-1}\right)}^{-1},$$
(17)$$\mu_{s_{t}}^{\left[m\right]}={\hat{s}}_{t}^{\left[m\right]}+\Sigma_{s_{t}}^{\left[m\right]}G_{s}^{T}Q_{t}^{[m]-1}\left(z_{t}-{\hat{z}}_{t}^{\left[m\right]}\right),$$
where $Q_{t}^{\left[m\right]}=G_{\theta }\Sigma_{n_{t},t-1}^{\left[m\right]}G_{\theta }^{T}+R_{t}$ is the covariance matrix of landmark observation, and $z_{t}$ is the actual observation of landmark location.
(18)$${\hat{z}}_{t}^{\left[m\right]}=g\left(\mu_{n_{t},t-1}^{\left[m\right]},{\hat{s}}_{t}^{\left[m\right]}\right)=∥\mu_{n_{t},t-1}^{\left[m\right]}-{\hat{s}}_{t}^{\left[m\right]}∥.$$

It is known from Equation (Equation 18) that the algorithm measures the distance between the user and the landmark.

#### 5.2.2. Map Update

The map update is updating the location of the current detected landmark. The algorithm updates the estimate of the observed landmark according to Equation (Equation 19),
(19)$$p\left(\theta_{n_{t}}|s^{t,[m]},n^{t},z^{t}\right)=\eta^{\left[m\right]}\underset{z_{t}∼N\left(g\left(\theta_{n_{t}},s_{t}^{\left[m\right]}\right),R_{t}\right)}{\underset{︸}{p(z_{t}|\theta_{n_{t}},s_{t}^{\left[m\right]},n_{t})}}\underset{\theta_{n_{t}}∼N\left(\mu_{n_{t},t-1}^{\left[m\right]},\Sigma_{n_{t},t-1}^{\left[m\right]}\right)}{\underset{︸}{p(\theta_{n_{t}}|s^{t-1,[m]},z^{t-1},n^{t-1})}}.$$

Recall that *g* is linearized to retain Gaussian distribution of the posterior. This leads to the following update Equations (Equation 20)–(Equation 22). The derivation of them is equal to that of the EKF measure update:(20)$$K_{t}^{\left[m\right]}=\Sigma_{n_{t},t-1}^{\left[m\right]}G_{\theta }^{T}Q_{t}^{[m]-1},$$
(21)$$\mu_{n_{t},t}^{\left[m\right]}=\mu_{n_{t},t-1}^{\left[m\right]}+K_{t}^{\left[m\right]}\left(z_{t}-{\hat{z}}_{t}^{\left[m\right]}\right),$$
(22)$$\Sigma_{n_{t},t}^{\left[m\right]}=\left(I-K_{t}^{\left[m\right]}G_{\theta }\right)\Sigma_{n_{t},t-1}^{\left[m\right]}.$$

The algorithm inherits the properties of FastSLAM and there are many possible landmark hypotheses, each of which corresponds to an individual particle which has its own judgment about the landmark, and this kind of multi-estimation attribute offers SLAM robustness to errors.

#### 5.2.3. Resampling

As the particles generated may not match the desired posterior, it is necessary to resample to remove the particles with great errors and put more in the right area. The reason that some particles generated do not yet match the desired posterior is η in Equation (Equation 19), which may be different for different particle particle. The normalizer is the inverse of the probability of the measurement: $\eta^{\left[m\right]}=p{\left(z_{t}|s^{t-1,[m]},u^{t},z^{t-1},n^{t}\right)}^{-1}$. To account for this mismatch, the algorithm resamples in proportion to the following importance factor: (23)$$w_{t}^{\left[m\right]}\propto p\left(z_{t}|s^{t-1,[m]},u^{t},z^{t-1},n^{t}\right)=\;\int \int \;\underset{z_{t}∼N\left(g\left(\theta_{n_{t}},s_{t}\right),R_{t}\right)}{\underset{︸}{p(z_{t}|\theta_{n_{t}},s_{t},n_{t})}}\underset{\theta ∼N(\mu_{n_{t},t-1}^{\left[m\right]},\Sigma_{n_{t},t-1}^{\left[m\right]})}{\underset{︸}{p(\theta_{n_{t}}|s^{t-1,[m]},u^{t-1},z^{t-1},n^{t-1})}}d\theta_{n_{t}}\underset{s_{t}∼N\left({\hat{s}}_{t-1}^{\left[m\right]},P_{t}\right)}{\underset{︸}{p(s_{t}|s_{t-1}^{\left[m\right]},u_{t})}}ds_{t}.$$

The distribution in Equation (Equation 23) can be approximated as a Gaussian distribution by linearizing *g* with the mean ${\hat{z}}_{t}$ and covariance $G_{s}P_{t}G_{s}^{T}+G_{\theta }\Sigma_{n_{t},t-1}^{\left[m\right]}G_{\theta }^{T}+R_{t}$.

In addition, the algorithm selects the data association $n_{t}$ satisfying Equation (Equation 24):(24)$${\hat{n}}_{t}^{\left[m\right]}=\underset{n_{t}}{\arg\max}p\left(z_{t}|n_{t},n^{t-1,[m]},s^{t,[m]},z^{t-1},u^{t}\right).$$

The distribution in Equation (Equation 24) is calculated as follows:(25)$$p\left(z_{t}|n_{t},n^{t-1,[m]},s^{t,[m]},z^{t-1},u^{t}\right)=\int \underset{z_{t}∼N\left(g\left(\theta_{n_{t}},s_{t}^{\left[m\right]}\right),R_{t}\right)}{\underset{︸}{p(z_{t}|\theta_{n_{t}},n_{t},s_{t}^{\left[m\right]})}}\underset{\theta_{n_{t}}∼N\left(\mu_{n_{t},t-1}^{\left[m\right]},\Sigma_{n_{t},t-1}^{\left[m\right]}\right)}{\underset{︸}{p(\theta_{n_{t}}|n^{t-1,[m]},s^{t-1,[m]},z^{t-1})}}.$$

It can be seen that linearization of *g* generate a Gaussian distribution over $z_{t}$ with mean $g(\mu_{n_{t},t-1}^{\left[m\right]},s_{t}^{\left[m\right]})$ and covariance $Q_{t}^{\left[m\right]}$, both of which are functions of the data assocaition variable $n_{t}$.

## 6. Results and Discussion

### 6.1. Experiment Introduction

This section evaluates the performance of UIP. Installing a UIP application on the phone and granting the required permissions so that it can grab sensor data from the phone, then send them to the server over the network. The system is implemented on an Android phone. When users walk, the sensors on the phone will get his pose information. The data is sampled (gyroscope at highest permissible rate; other sensors at 24 Hz) and then they are sent to the server for processing. The server side code is written using C♯ and MATLAB (MATLAB2017a, MathWorks, Natick, MA, USA), and all algorithms run on the server. When a new landmark is detected, the server will update the landmark list. Physics laboratory building is selected as the experimental platform, and all landmarks are decomposed into: eight turns, nine magnetic, 15 WiFi landmarks and three global landmarks. In the experiment, three users move around the building to collect data, and each user can use the landmarks detected by the previous users.

Figure 10 shows the plan of the experimental area and some landmarks within it. Everyone walks along the corridor from the entrance to the floor.

### 6.2. Landmark Detection

#### 6.2.1. Global Landmark

Figure 11 shows the false positive and false negative rates of different GLMs, and it can be seen that some landmarks are easier to detect than other landmarks as the unique patterns. Most values are zero in Figure 11, and the overall false positive rate and false negative rate are about 0.002 and 0.011, respectively, which means that the detection of GLMs is accurate.

#### 6.2.2. Local Landmark

Figure 12a,b show the location accuracy of the calculated landmark and the number of the observed landmarks change over time respectively. As more users explore the environment, the number of landmarks will increase over time, and the accuracy of the landmarks will also increase as different paths lead to different independent estimates. In about 2 h, the average positional accuracy can converge to less than 1 m.

Factually, data in the experiment is limited in the diversity of paths for the limitation of motion areas, and positioning accuracy may increase due to increased path diversity in practical applications. It can be seen from Figure 12c that the false positive rate detected by the three types of landmarks in the experiment is less than 1%, indicating that the matching accuracies are quite high. Similar to GLM, different LLMs have different detection accuracy due to different patterns, but each landmark has an accuracy of more than 60%.

#### 6.2.3. Landmark Quantity

Although there are a large number of landmarks in the experimental platform, it is difficult to learn how many landmarks exist in other environments, and whether there will be a large impact on positioning if there are fewer landmarks in some buildings. Figure 13 shows the positioning errors of different numbers of landmarks, and even though there are only 10 landmarks, the average positioning error is less than 2 m. In addition, landmarks based on more dimensions (e.g., acoustic signal) can be selected if needed.

### 6.3. UIP Positioning Assessment

#### 6.3.1. Factors Affecting Performance

This section evaluates the impact of different factors on the positioning performance, and the main indicator is the cumulative distribution function (CDF).

(1) Particle number: The comparison result is shown in Figure 14. It can be seen that the positioning accuracy of 50 particles is greatly improved compared with that of 25 particles; however, if the number of particles continues to increase, the positioning accuracy will not be significantly improved, and the amount of calculation will increase greatly. Therefore, the performance is saturated with about 50 particles where the median accuracy is about 0.53 m, and the number is used in subsequent experiments.

(2) Global Landmark: Figure 15 compares the CDF of positioning error with/without GLMs. In Figure 15, the average error still can be less than 1.5 m without GLMs, and in the case of using global landmarks, the performance is greatly improved, CDF within one meter increases by about two times, which is 0.8; the value is over 0.9 within 2 m, which increased by 0.1, indicating that the GLM is a successful idea.

(3) Error over time: The Euclidean distance error of the system is calculated to evaluate the positioning accuracy, and Figure 16 shows the variation of the error over time, which is undulating. Due to noise in the sensor readings, the error increases as the user walks, and it will decline when a landmark is observed. As more landmarks are observed, map accuracy increases and user positioning errors decrease, hence the overall trend of the curve is decreasing. The average positioning error does not exceed one meter, which is enough to satisfy most indoor positioning tasks.

(4) Online and offline: Figure 17 shows the error distributions in two patterns. In offline patterns, the system learns its error when a user encounters a landmark, so it can track and partially correct past traces, which is not available in online mode. The figure shows that offline system can observably improve the smearing of distribution in online positioning, and the median distance error is slightly enhanced too. Moreover, although the positioning accuracy of the online mode is not as good as offline mode, its error is basically controlled within 1 m.

#### 6.3.2. SLAM Framework

In order to evaluate the SLAM framework in Chapter 4, the systems with/without the SLAM framework are compared in convergence time and positioning accuracy. Uncertainty of landmarks or user locations is considered in the SLAM system, which is not available in the non-SLAM system.

(1) Convergence time: It is the time from the beginning to the point where the positioning accuracy tends to be stable. The length of convergence time affects the positioning efficiency of the system. Figure 18 shows that the accuracy of the SLAM system converges quickly within 1 h and eventually achieves an average accuracy of approximately 0.83 m, and it is better than the non-SLAM system, which converges to an average accuracy of about 1 meter after 2 h.

(2) Error distribution: Figure 19 compares the error distribution of the systems with/without SLAM. It can be seen that, after using the SLAM framework, the positioning accuracy is improved significantly, and the median positioning error goes down by about one meter. In the SLAM system, CDF of the positioning error within one meter is about 0.8, while it is about 0.3 in the non-SLAM system. The error of system with SLAM is basically controlled within two m, it can significantly improve the accuracy. As can been seen from Figure 18 and Figure 19, the SLAM system can obtain a high positioning accuracy in a short period of time.

#### 6.3.3. Other Systems

The UIP system is compared with two classical systems: the WiFi fingerprint identification system [32] and the magnetic fingerprint recognition system [33]. Figure 20 shows that the UIP has a smaller median positioning error which is about 0.5 m, while it is about 0.7 m in the WiFi system and it exceeds one meter in the magnetic system. In addition, CDF of the localization error in UIP exceeds 0.8 within one meter and 0.9 within 1.5 m, which has an obvious advantage over other systems. However, they have better worst-case errors due to the initial manual calibration, and the distribution has a shorter tail.

## 7. Conclusions

In order to avoid the complicated calibration of indoor positioning, an unsupervised indoor positioning (UIP) system is proposed in the paper, which uses landmarks to compensate for the error of dead reckoning. The dead reckoning and landmark observation data are input into the SLAM algorithm to realize online positioning of users.This paper validates the feasibility of the idea of landmarks and provides detailed descriptions of the landmarks and the SLAM framework. The experiments are carried out in many aspects, showing that UIP can find different landmarks accurately, and the probability of error identification is less than 1%. In addition, the positioning error can converge quickly, eventually achieving a median positioning error of 0.53 m, which is better in the offline mode. Compared with the two classical algorithms, UIP also has obvious advantages in positioning accuracy. Future works will be expanding the system, including coordinated positioning of multiple people, using other sensors such as acoustic sensors and optical sensors.

## Figures and Tables

**Figure 1 entropy-21-00327-f001:**
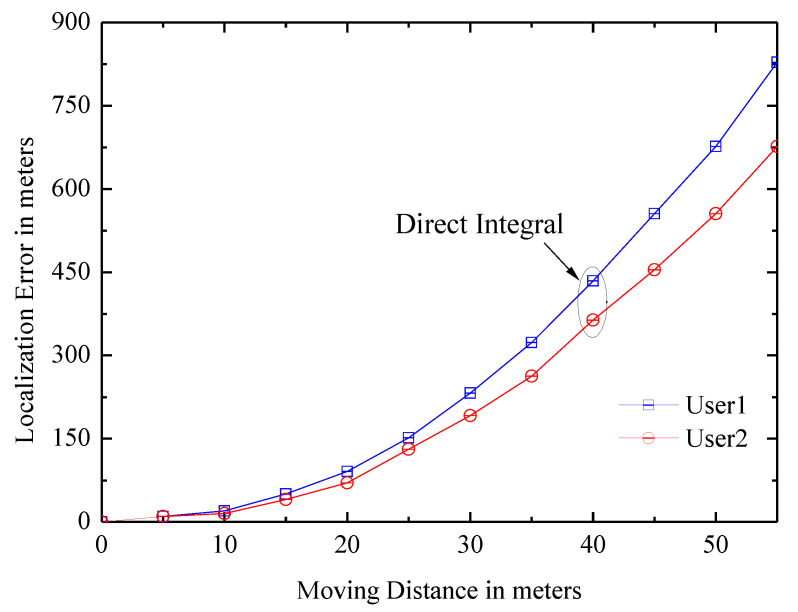
The positioning error is quickly accumulated with the movement.

**Figure 2 entropy-21-00327-f002:**
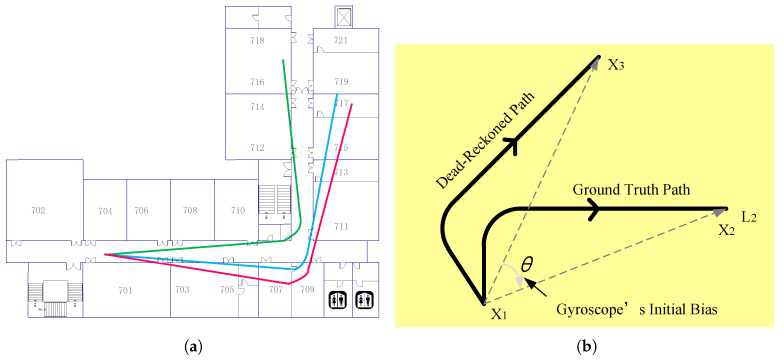
(**a**) path estimates; (**b**) using the landmark to correct the deviation.

**Figure 3 entropy-21-00327-f003:**
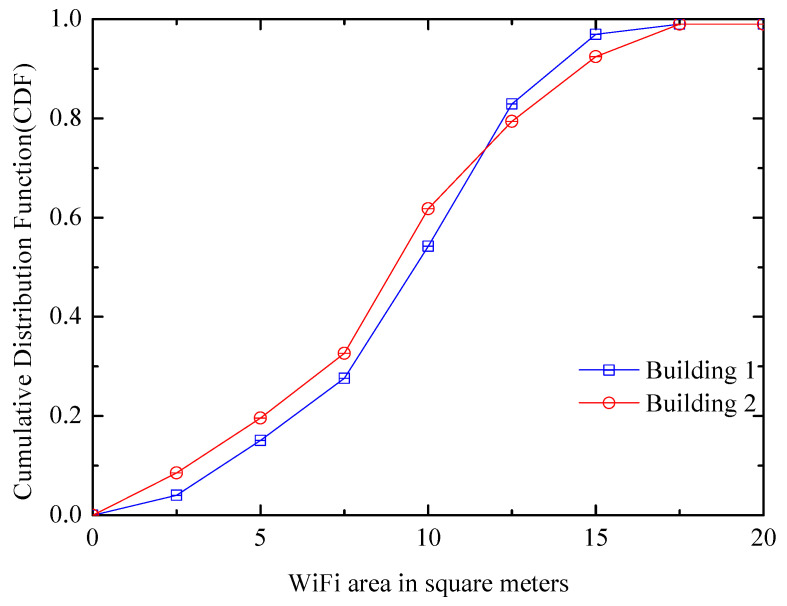
WiFi area distribution.

**Figure 4 entropy-21-00327-f004:**
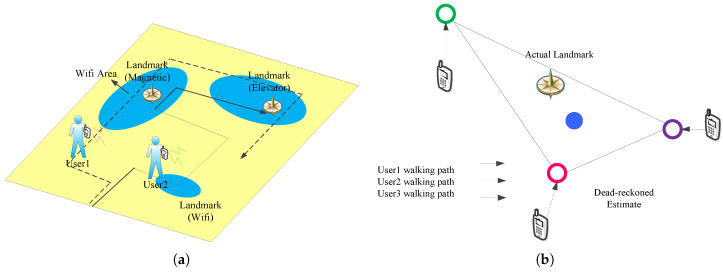
(**a**) landmarks are encountered while walking; (**b**) the estimation of landmark location.

**Figure 5 entropy-21-00327-f005:**
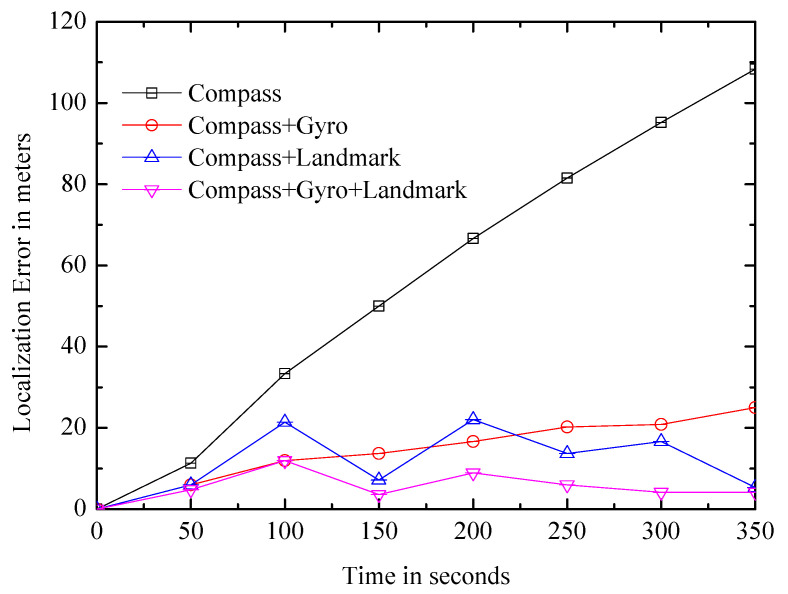
Comparison of positioning errors.

**Figure 6 entropy-21-00327-f006:**
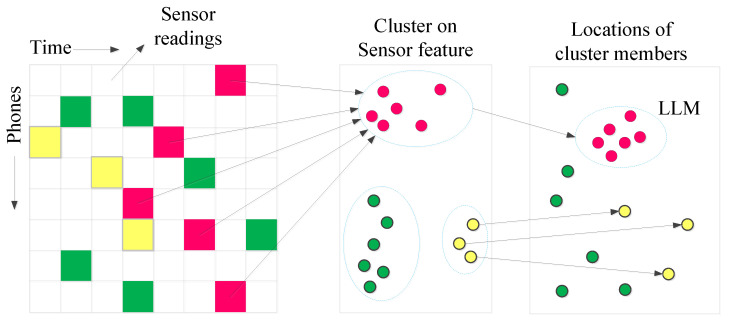
The process of data collection and classification.

**Figure 7 entropy-21-00327-f007:**
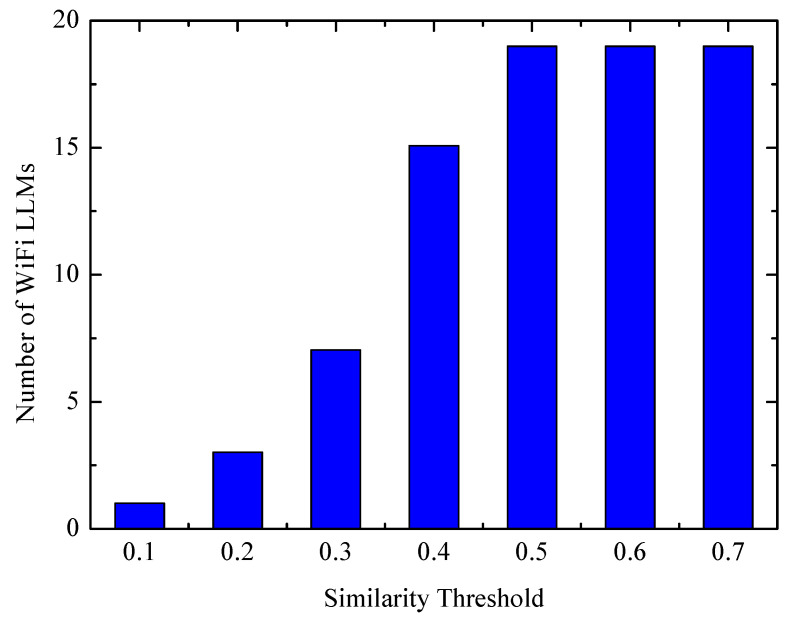
The landmark number over the threshold.

**Figure 8 entropy-21-00327-f008:**
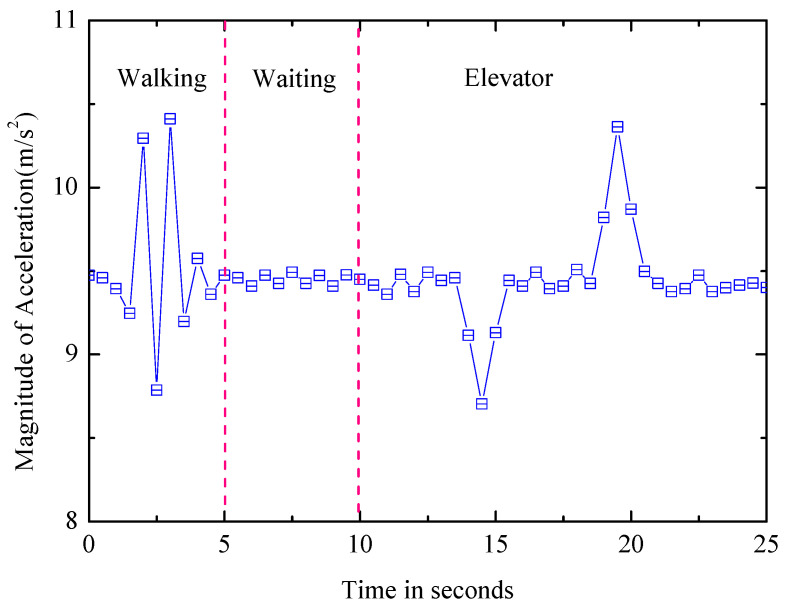
Process of accelerometer signal change.

**Figure 9 entropy-21-00327-f009:**
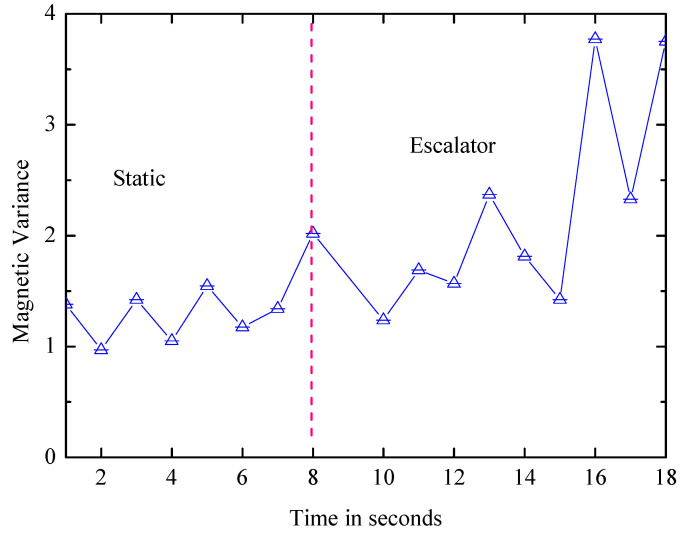
Magnetic difference between the static and escalator.

**Figure 10 entropy-21-00327-f010:**
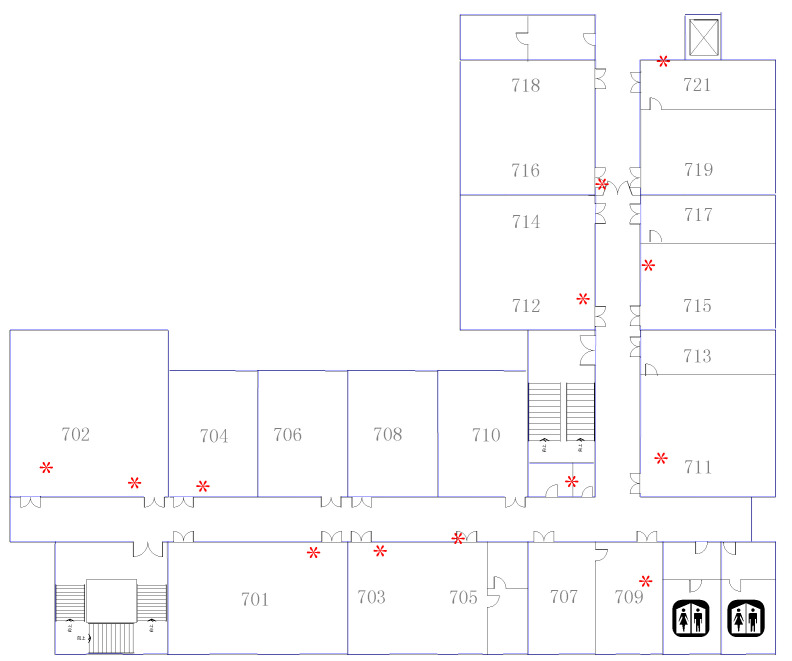
Experimental area plan in 54 m × 42.6 m.

**Figure 11 entropy-21-00327-f011:**
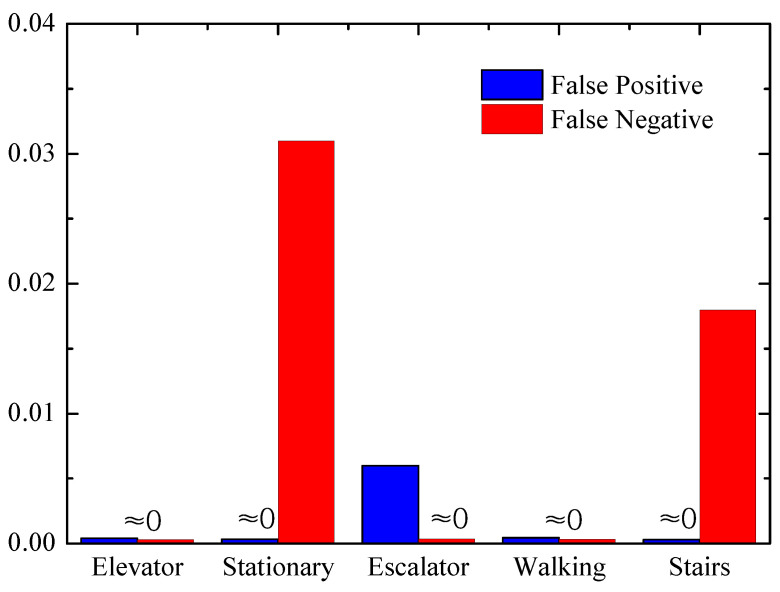
Global landmark recognition.

**Figure 12 entropy-21-00327-f012:**
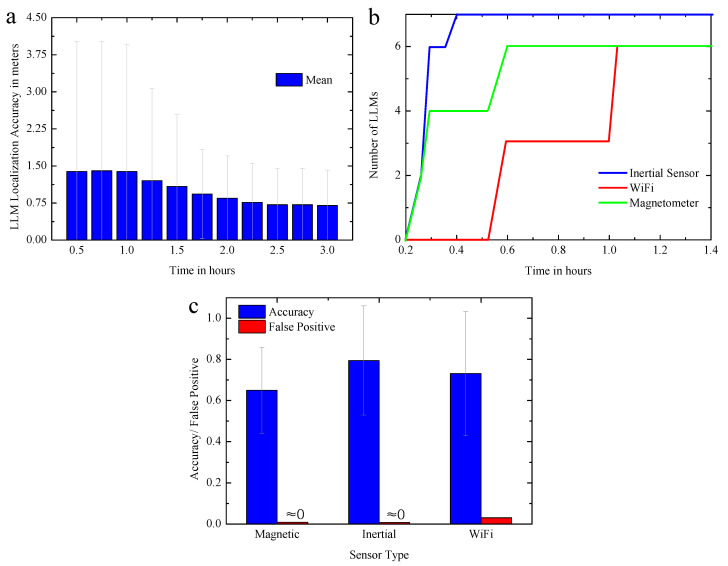
(**a**) landmark positioning accuracy over time. (**b**) the number of landmarks discovered over time; (**c**) recognition rate of different landmarks.

**Figure 13 entropy-21-00327-f013:**
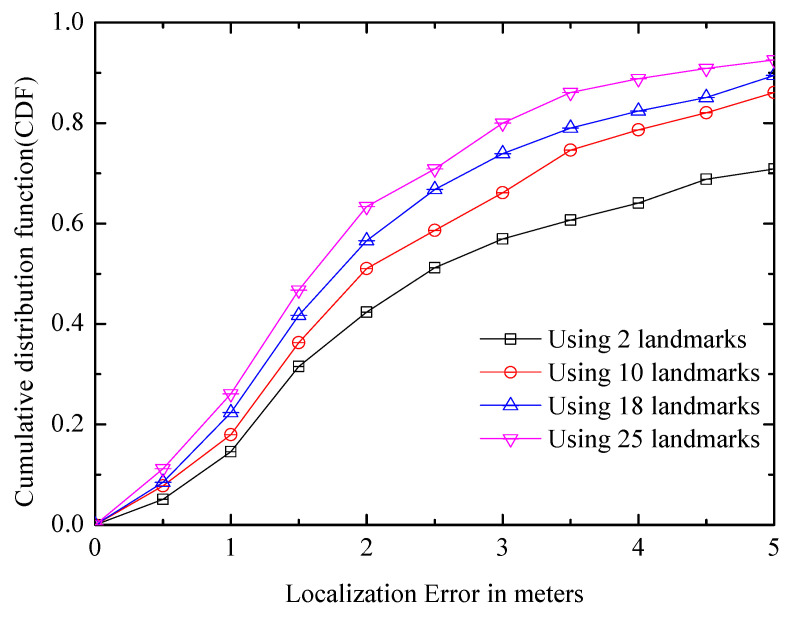
Comparison of systems with different landmark numbers.

**Figure 14 entropy-21-00327-f014:**
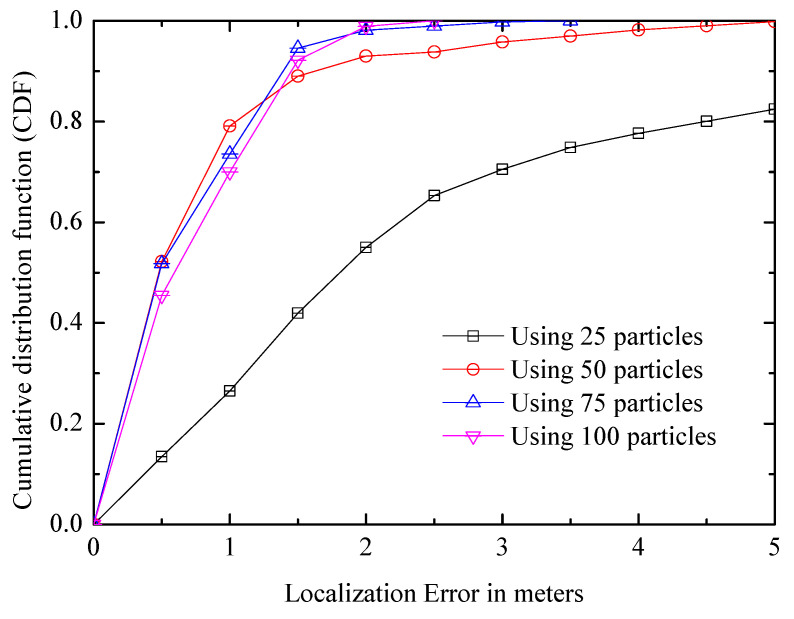
Comparison of systems with different particle numbers.

**Figure 15 entropy-21-00327-f015:**
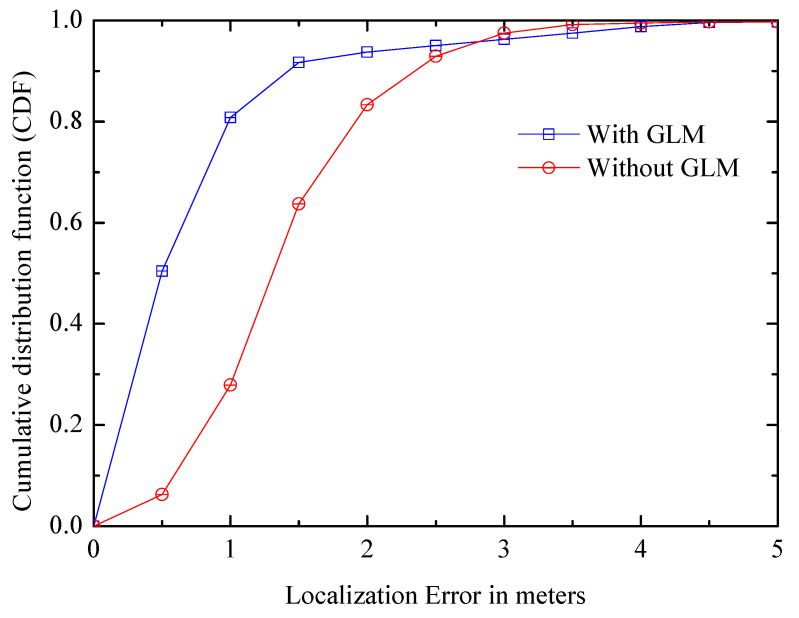
Comparison of systems with/without global landmarks.

**Figure 16 entropy-21-00327-f016:**
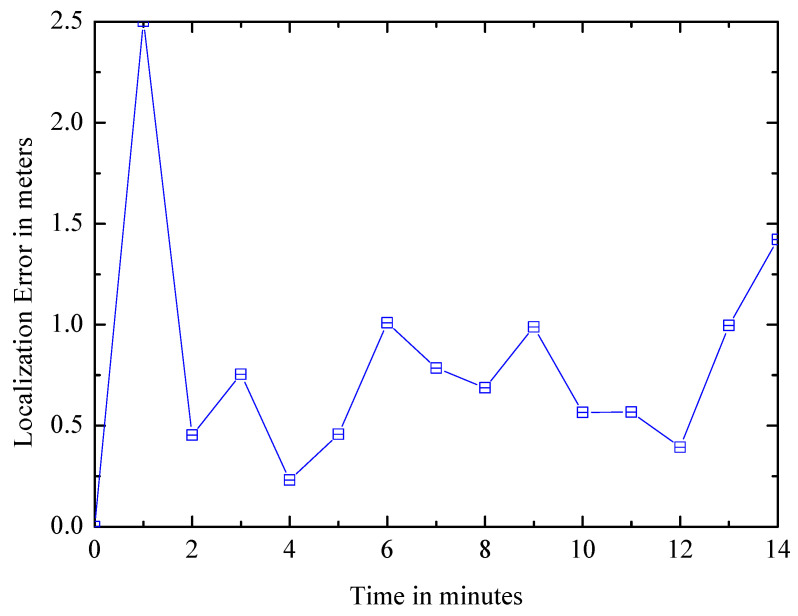
Positioning error over time.

**Figure 17 entropy-21-00327-f017:**
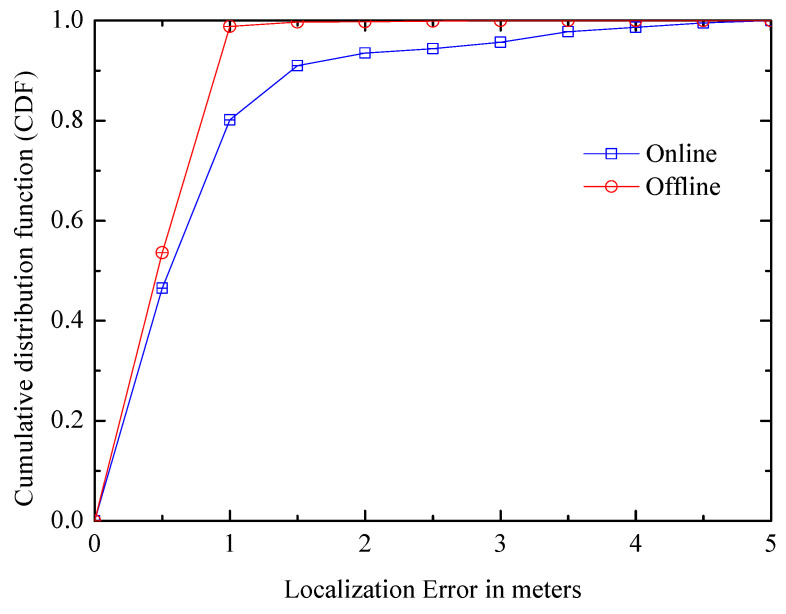
Comparison of online/offline systems.

**Figure 18 entropy-21-00327-f018:**
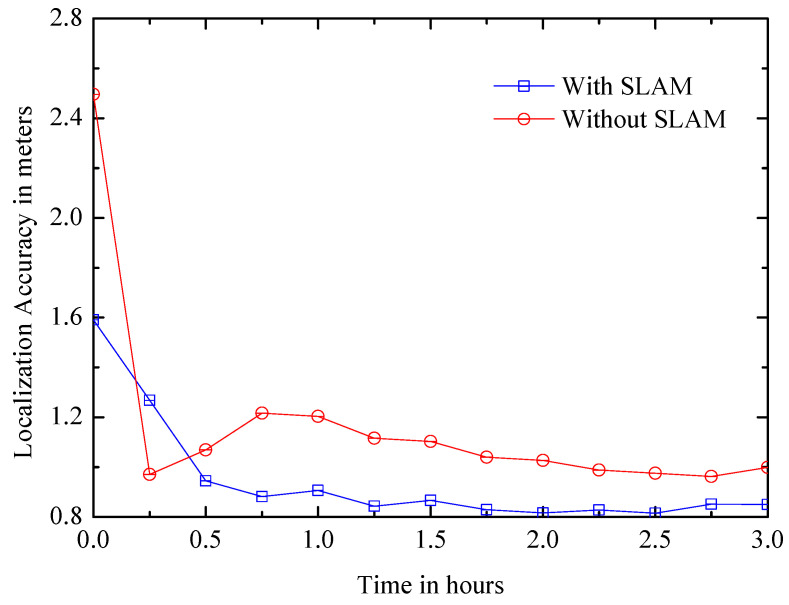
Positioning accuracy over time.

**Figure 19 entropy-21-00327-f019:**
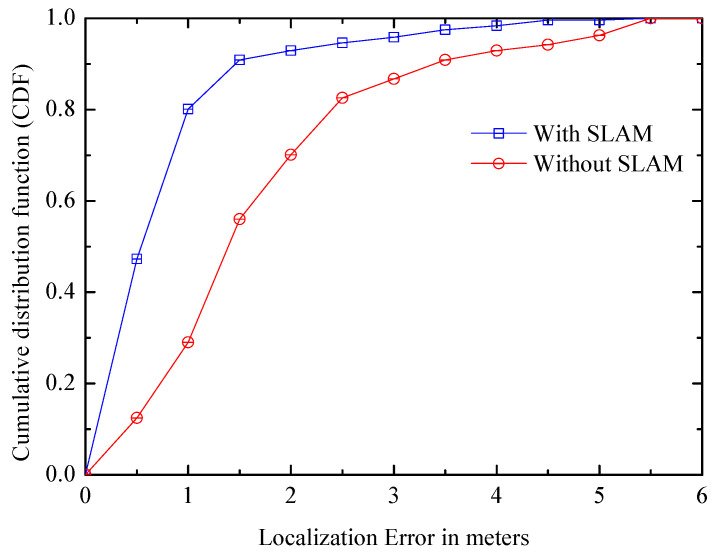
Positioning error distributions.

**Figure 20 entropy-21-00327-f020:**
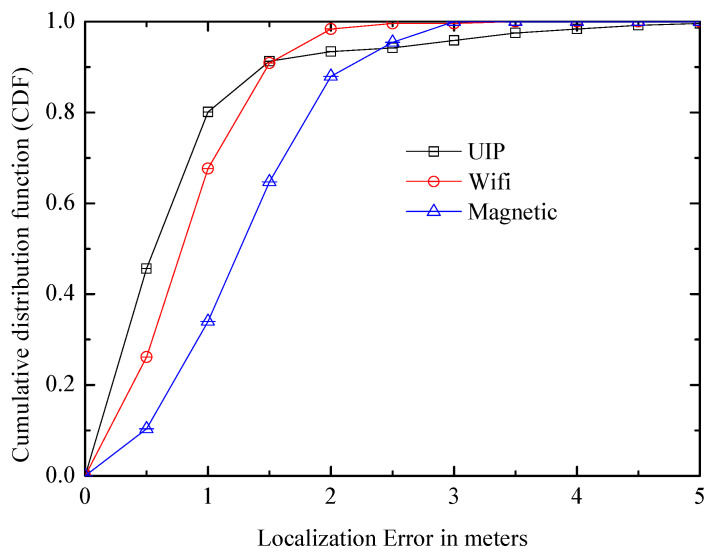
Comparison of unsupervised indoor positioning(UIP) with two other systems.

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
