# Peer review of "Unsupervised Indoor Positioning System Based on Environmental Signatures"

_entropy, 2019, doi:10.3390/e21030327_

Round 1

Reviewer 1 Report

Fig 3
< time, displacement, direction>
How is the displacement obtained (e.g. Step detector or accelerometer double integration).
Why is the baseline curve linear ? Acceleration offset would lead to a quadratic function over time.
Line 86 states that the average error would be 1.2 meter, but fig. 3 indicates an average error in the pink curve of about 5 meter.

Figure 4 is unclear. What error is measured ?

Fig. 5 In literature, magnetic landmarks are typically defined as a magnetic pattern over a certain path, and the correlation between the real path and the magnetic fingerprint is by the so-called time warp algorithm. Taking just absolute readings at a certain position for marking to me does not make much sense, as the same value typically exists at a variety of places.
Is only the absolute value of the magnetic field used or is the absolute value and the z-coordinate used ?

line 159 states that a not received RSSI value would be regardes as zero. Usually logarithmic units are used, and a value of 0dBm would represent a very stron signal. Please state in what units RSSI is used here and why it is useful to use the value 0 for an unreceived AP

Chapter 3.2.3 Resampling:
Typically, reasmpling uses information which is not included in the propagation calculation, e.g. map information or a route graph. It should be explained not only mathematically, but also conceptual, how the importance factor is derived and what physical meaning it has.

4.1. Experiment introduction
A map of the region under investigation has to be supplied.

Chapter 5
Unsupervised indoor positioning (UIP) systems, as is proposed in the paper, are a topic of intense reserarch and, for example at the ipin conference series, a lot of papers on this topic is published. This state of the art should be included in the introduction.

Author Response

Thanks for your valuable advice. The paper has been modified according to your suggestions. The modified part is highlighted in the PDF and responds to all suggestions in the Word document.

Reviewer 2 Report

The paper presents an unsupervised indoor positioning method that takes advantage of environment features in indoor environments and considers these signatures as landmarks and combines dead reckoning with them in a SLAM frame to reduce positioning errors and convergence.

My first concern is about the scope of the journal. The Aims & scope of the Entropy journal includes the following topics:

develop the theory behind entropy or information theory

provide new insights into entropy or information-theoretic concepts

demonstrate a novel use of entropy or information-theoretic concepts in an application

obtain new results using concepts of entropy or information theory

In my opinion, the paper cannot be included in any of the previous topics. If authors want to publish the actual work in this journal, they must emphasise in which aspect the proposed work improves some of the previous topics. Maybe MDPI Sensors will be more adequate for this paper.

About the paper: the idea is interesting but the explanation of the paper should be improved. It is very difficult to understand the proposed method.  The experiments section needs also more explanations.

Some comments at each section:

1) Introduction

“The dead reckoning approach based on sensors of smartphones has begun to receive attention [13] [14] in recent years”

I agree that dead reckoning approaches are relevant, but references [13] and [14] are not very recent. Authors should include some other more recent references.

Introduction section should end by explaining the main novelty of the paper with respect to other state-of-art works, i.e. why the proposed method is novel?

It is also a good idea to finish the introduction with a paragraph describing the organization of the rest of the paper.

2) Related work

There is not a related work section.

3) Mobile sensor positioning

4) Landmark design

5) SLAM

The problem of these three sections is that it is not clear if the authors are explaining how these techniques works or they are presenting this proposed method.

I guess that there is a mix of both things. I suggest a section “background” performing a brief introduction to the well-known techniques and other section explaining the proposed method.

Line 43: About dead reckoning, how your system estimates the first location to start the process?

Line 56:  “The paper uses UPTIME[20]” Which paper? I guess that it is the paper presented by the authors. Why not explaining the method presented in [20] if it is the one used by the authors?

In general, figures caption should show more information. For instance, Figure 1 is an example of a caption not providing useful information.

Section 1.1.2 Test: this is not the place to perform experiments.

Sections 1.2 and 2. The most important part of the paper is the way to obtain the landmarks. Unfortunately, the explanation provided in the paper is very confusing. Authors should improve the explanation. In particular, how local landmarks are obtained.

Section 3. SLAM is a well-known technique. There is too much detail in the explanation. Authors should provide a short introduction and explain in detail how this technique is used in their method.

6) Results and discussion

Similar problems happen in this section. Authors should clarify how the data has been acquired.

Section 4.3.3 is crucial to judge if the method is novel or not. Authors should emphasise this explanation. It is important to provide a short description of the methods presented in [34] and [35] and which are the improvements of the proposed method with respect to these ones (for instance, this can be done in the related work section).

In general, the idea seems to be a good idea, but the explanations and organization of the paper should be improved to be accepted to be published. In addition, I suggest changing to another journal with a more related scope.

Author Response

(The authors gave the same response as above.)

Round 2

Reviewer 1 Report

Figure 3: the X-Axis is described as diagonal length in meters, while in the explaining text it says that x is the coverage area measured in square meters

Figure 10: The caption should indicate the size of the map (e.g. something like 10m x 20m, numbers to be replaced by the real numbers)

Author Response

Thanks for your valuable comments, I have modified the relevant content.

Reviewer 2 Report

My opinion about the paper is more or less the same than when the previous version.

Some important problems of the paper are as follows:

1) I think that the paper is out of the scope of the journal. There are other journals (as for instance MDPI sensors) within the scope of the paper. One of my comments in the previous version (and I think that it was one of the most important) was: “If authors want to publish the actual work in this journal, they must emphasise in which aspect the proposed work improves some of the previous topics”. This is missing in the next version.

2) Revise the paper since there are too many typos

3) Introduction

The introduction should clearly explain the main ideas of the method and should expose why it is novel. Both are missing. This is also very important to judge the proposed method.

Some state of art methods are not well explained: “[3] also proposed a noval method”

4) Related work

The related work section can be improved adding more state of arts methods.

Authors have explained the Horus and MaLoc algorithms, but which are their problems? Why the proposed method is preferable to them?

5) Section 2 should be rewritten. I guess that authors are performing something similar to a background section.

6) Section 3. Authors should explain better how the proposed method works.

Author Response

Thanks for your valuable suggestions, I am very sorry for the result of the first modification. We have organized the recommendations and responded one by one.
